# Gravitational clustering of cosmic relic neutrinos in the Milky Way

Jue Zhang[1] & Xin Zhang [2]

The standard model of cosmology predicts the existence of cosmic neutrino background in the present Universe. To detect cosmic relic neutrinos in the vicinity of the Earth, it is necessary to evaluate the gravitational clustering effects on relic neutrinos in the Milky Way. Here we introduce a reweighting technique in the N-one-body simulation method, so that a single simulation can yield neutrino density profiles for different neutrino masses and phase space distributions. In light of current experimental results that favor small neutrino masses, the neutrino number density contrast around the Earth is found to be almost proportional to the square of neutrino mass. The density contrast-mass relation and the reweighting technique are useful for studying the phenomenology associated with the future detection of the cosmic neutrino background.

[1] Center for High Energy Physics, Peking University, Beijing 100871, China. [2] Department of Physics, College of Sciences, Northeastern University, Shenyang 110819, China. Correspondence and requests for materials should be addressed to X.Z. (email: zhangxin@mail.neu.edu.cn)

The standard model of cosmology predicts that neutrinos were decoupled from the thermal bath when the temperature of the Universe was about 1 MeV. These relic neutrinos constitute the current cosmic neutrino background. Detecting cosmic relic neutrinos[1], as proposed in the upcoming PTOLEMY experiment[2], is thus a direct test of the standard model of cosmology, and can push our understanding of the Universe to its age of about one second. In order to detect them in the neighborhood of the Earth, a prerequisite would be to figure out the number density of relic neutrinos at our local environment. Although the standard model of cosmology does predict that the average number density of relic neutrinos in the current Universe is about 56 cm$^{-3}$ for each flavor[3], more relic neutrinos can be accreted around the Earth, due to the fact that massive neutrinos suffer from the gravitational potential of both dark matter (DM) and baryonic matter in the Milky Way (MW). Investigating the gravitational clustering of relic neutrinos is thus a necessary step towards interpreting the results from the future detection of cosmic neutrino background.

Gravitational clustering effects are often studied numerically with the N-body simulation method. However, to reach a resolution of ~8 kpc, the distance from the Earth to the galactic center of the MW, the N-body simulation turns out to be computationally expensive[4]. In 2004, a restricted but effective method called N-one-body simulation was proposed to evaluate the gravitational clustering effects of relic neutrinos[5]. In contrast with the N-body simulation, where all the interactions among particles are included, relic neutrinos in the N-one-body simulation are assumed to evolve under the gravitational potential of both DM and baryonic matter. The back reaction, i.e., the gravitational effects of neutrinos on the clustering of DM and baryonic matter, and the gravitational interactions among neutrinos are both considered to be negligible[5,6]. This assumption works for the evolution of the Universe at a late stage ($z \lesssim 3$ with $z$ being the redshift), when the energy density of neutrinos is much smaller than that of DM[3]. To implement the N-one-body simulation, one first divides the initial phase space of neutrinos into $N$-independent chunks, and then evolves each chunk following a one-body motion in the gravitational potential generated by DM and baryonic matter. Assembling all the $N$ chunks with their corresponding weights after the evolution yields the final phase space distribution of neutrinos.

In this work we introduce a reweighting technique in the N-one-body simulation, so that a single N-one-body simulation is sufficient to yield neutrino density profiles for different neutrino masses and phase space distributions. For small neutrino masses, we find that the neutrino number density contrast is almost proportional to the square of neutrino mass. The dependence of gravitational clustering effects on the phase space distribution is also investigated, followed by the implications of gravitational clustering effects on interpreting the results from the future detection of cosmic neutrino background.

## Results

**Normalized evolution equations**. Here we adopt a generalized Navarro–Frenk–White (NFW) profile[7] for the DM distribution in the MW, while for the baryonic matter distribution a spherically symmetric profile is also assumed for simplicity[8]. See the Methods section for the details about the matter density profiles used in the numerical simulation.

Within the spherical gravitational potential $\phi(r)$, the one-body motion of a test particle with the mass $m_\nu$ is confined to a plane, and obeys the following Hamiltonian equations[9]

$$\frac{dr}{d\tau} = \frac{p_r}{am_\nu}, \quad \frac{dp_r}{d\tau} = \frac{\ell^2}{am_\nu r^3} - am_\nu \frac{\partial\phi}{\partial r}, \quad (1)$$

where $a = 1/(1 + z)$ is the scale factor of the Universe, $\tau$ is the conformal time defined as $d\tau = dt/a(t)$, and $p_r = am_\nu \dot{r}$ and $\ell = am_\nu r^2 \dot{\theta}$ are the canonical momenta conjugate to $r$ and $\theta$, respectively. Here the dot denotes the derivative with respect to $\tau$, and $(r, \theta)$ are the polar coordinates in the comoving frame. The gravitational potential $\phi(r, \tau)$ also evolves and its evolution is assumed to be independent of relic neutrinos in the N-one-body simulation. Because of spherical symmetry, $\ell$ is a conserved quantity and we may ignore the motion in the $\theta$ direction. A key observation in developing the reweighting technique is to identify that the evolution equations in Eq. (1) can be written in a form independent of the neutrino mass $m_\nu$. Namely, with the normalized quantities $u_r \equiv p_r/m_\nu = a\dot{r}$ and $u_\theta \equiv \ell/m_\nu = ar^2\dot{\theta}$, the following normalized evolution equations can be derived[5]

$$\frac{dr}{dz} = -\frac{u_r}{da/dt}, \quad \frac{du_r}{dz} = -\frac{1}{da/dt}\left(\frac{u_\theta^2}{r^3} - a^2\frac{\partial\phi}{\partial r}\right). \quad (2)$$

These normalized evolution equations are the ones implemented in our N-one-body simulation.

**Reweighting technique**. The essence of the reweighting technique is to let a test particle represent all the particles within a fixed interval $(r_a, u_{r,a}, u_{\theta,a}) \rightarrow (r_b, u_{r,b}, u_{\theta,b})$ in the space spanned by $(r, u_r, u_\theta)$. The size of the interval does not depend on the mass or the phase space distribution of relic neutrinos. The dependences on these quantities arise when associating weight to the fixed interval $(r_a, u_{r,a}, u_{\theta,a}) \rightarrow (r_b, u_{r,b}, u_{\theta,b})$. To illustrate that, we first introduce another variable set of $(r, y, \psi)$, with $y = p/T_{\nu,0}$. Here $p$ denotes the magnitude of the canonical momentum, $\psi$ is the direction of momentum with respect to the positive radial direction, and $T_{\nu,0}$ is the neutrino temperature at the present time. The transformations between $(u_r, u_\theta)$ and $(y, \psi)$ are given by $\psi = \tan^{-1}(ru_r/u_\theta)$ and $y = m_\nu u_r/(\cos\psi T_{\nu,0})$. Therefore, the fixed interval $(r_a, u_{r,a}, u_{\theta,a}) \rightarrow (r_b, u_{r,b}, u_{\theta,b})$ corresponds to an interval $(r_a, y_a, \psi_a) \rightarrow (r_b, y_b, \psi_b)$, which will have varying lower limit $(y_a)$ and upper limit $(y_b)$ depending on the neutrino mass $m_\nu$. For the phase space interval $(r_a, y_a, \psi_a) \rightarrow (r_b, y_b, \psi_b)$, we obtain its associated weight $dw$ as follows[5]

$$dw = 8\pi^2 T_{\nu,0}^3 \int_{r_a}^{r_b} r^2\,dr \int_{y_a}^{y_b} f(y)y^2\,dy \int_{\psi_a}^{\psi_b} \sin\psi\,d\psi, \quad (3)$$

where spherical symmetry is applied, and $f(y)$ is the phase space distribution function. In the case of thermal relic neutrinos, $f(y)$ follows the Fermi–Dirac form

$$f^{FD}(y) = \frac{1}{1 + e^y}. \quad (4)$$

The effect of neutrino masses reflects then in the lower and upper limits of $y$ for a fixed interval in terms of $(r, u_r, u_\theta)$, while for different phase space distributions one simply uses the corresponding forms of $f(y)$.

In practice, one still needs to perform a benchmark simulation with definite neutrino mass and phase space distribution. This benchmark simulation serves two purposes. First, the one-body evolutions of $N$ test particles can be obtained. Second, in the benchmark simulation the initial phase space of relic neutrinos is discretized, and such a discretization would fix the interval $(r_a, u_{r,a}, u_{\theta,a}) \rightarrow (r_b, u_{r,b}, u_{\theta,b})$ for each evolved sample. When switching to another neutrino mass or a different phase space distribution, on one hand we can reuse those one-body evolution results from the benchmark simulation, and on the other hand we can associate a new weight to each evolved sample according to Eq. (3). With this reweighting technique, we then do not need to rerun the N-one-body simulation for different neutrino masses

and phase space distributions, so that lots of computing time can be saved.

**Thermal relic neutrinos.** In this work we consider a benchmark N-one-body simulation with $m_\nu = 0.15$ eV and thermal Fermi–Dirac phase space distribution. See the Methods section for the details about this benchmark simulation. The reweighting technique then enables us to obtain neutrino densitiy profiles $n_\nu(r)$ for other neutrino masses and phase space distributions. Consider the thermal phase space distribution first. In Fig. 1a, we show the neutrino density contrast $\delta_\nu$, defined as $\delta_\nu \equiv n_\nu/\bar{n}_\nu - 1$ with $\bar{n}_\nu$ being the average neutrino number density ($\delta_\nu$ corresponds to $f_c - 1$ in ref. [8]), as a function of the distance $r$ to the galactic center of MW. The results of four different neutrino masses are shown, and the neutrino halos can extend up to a few mega parsecs. At the location of the Earth ($r_\oplus = 8$ kpc) the neutrino density is enhanced by about 10% (115%) for the case of $m_\nu = 0.05$ (0.15) eV, due to the gravitational clustering effects.

For a fixed distance $r$, the relationship between the neutrino density contrast $\delta_\nu$ and $m_\nu$ is displayed in Fig. 1b. We observe that for the three different distances of $r$, all the scatter points obtained from the N-one-body simulation can be well fitted by a power-law function of $\delta_\nu \propto m_\nu^\gamma$. The obtained exponents $\gamma$ are around two for all cases, indicating that the linear approximation[5,6] is appropriate in light of current cosmological constraints that favor small neutrino masses[10]. Recall that in the Vlasov equation[9] for the phase space distribution function there exists a term involving both the gravitational potential $\phi$ and the distribution function $f$. In the linear approximation, one approximates the distribution function $f$ in that term with the corresponding distribution function $f_0$ without the presence of gravitational potential, so that the modified Vlasov equation becomes linear in both $\phi$ and $f$. The underlying requirement for making the linear approximation is that the perturbed distribution function $f$ should be close to the unperturbed one $f_0$, or the neutrino density contrast does not exceed greatly over order unity. For the neutrino masses considered in this work, according to Fig. 1 we find that the gravitational clustering effects are moderate so that the linear approximation works well here. However, if larger neutrino masses or heavier halo masses were considered, because of more enhanced gravitational clustering effects, the linear approximation would no longer be applicable, and the resulting power-law indices could have large deviations from two[5].

At the location of the Earth, the fitted power-law function for thermal relic neutrinos is given by

$$\delta_\nu^{\mathrm{FD}}(r_\oplus) = 76.5\left(\frac{m_\nu}{\mathrm{eV}}\right)^{2.21}, m_\nu \in [0.04, 0.15] \text{ eV}. \quad (5)$$

A similar power-law function[11] was obtained for higher neutrino masses $m_\nu \in [0.15, 0.6]$ eV, by fitting previous N-one-body simulation results[5]. In the recent literature two benchmark values of $m_\nu = 0.06$ eV and 0.15 eV are also studied[8]. Since the adopted DM and baryonic matter profiles in this work are the same as those in ref. [8], we can directly compare our results with those in ref. [8]. After taking into account the uncertainties from discrete sampling, we find that both results agree with each other, validating the reweighting technique. With the above fitted relation we can obtain neutrino number densities for all neutrinos within the mass range of [0.04, 0.15] eV. For $m_\nu < 0.04$ eV, the clustering effects due to the gravitational potential of baryonic matter and DM in the MW are insignificant, namely, the neutrino density contrast is less than about 0.05 at the location of the Earth. As a result, other astrophysical uncertainties, such as the contribution from the Virgo cluster[8,12], may play more important role in predicting the local neutrino number densities. Furthermore, when $m_\nu < 0.04$ eV, the required energy resolution $\Delta$ (full width at half maximum of the Gaussian distribution) is estimated to be $\Delta \simeq 0.7m_\nu < 0.03$ eV[13], which is beyond the reach of the current proposal of the PTOLEMY experiment[2]. For these reasons, we choose not to consider the neutrino masses below 0.04 eV here.

**New physics scenarios with non-thermal relic neutrinos.** It is also interesting to consider new physics (NP) scenarios, in which some chiral states of neutrinos in the early Universe are non-thermal and possess a phase space distribution that is significantly deviated from the thermal Fermi–Dirac distribution. For illustration, we consider a fully degenerate phase space distribution[14],

$$f^{\mathrm{deg}}(y) = 1, y < y_0, \quad (6)$$

where $y_0 = 1.76$ ensures the same average neutrino density as the thermal case. With the reweighting technique, we can also obtain the neutrino density profiles for this non-thermal case from the benchmark simulation. From Fig. 2, we observe that the neutrino contrast $\delta_\nu$ is about twice of that in the thermal case when $m_\nu \lesssim 0.1$ eV. This is due to the fact that in the fully degenerate case more relic neutrinos reside in the low momentum states. For

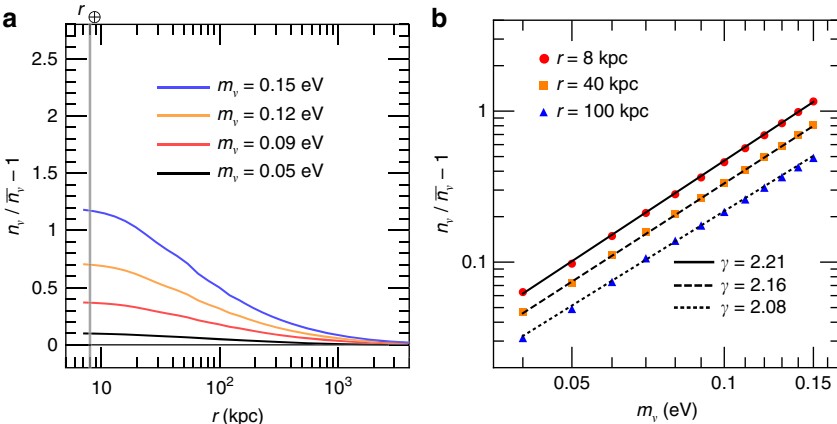

**Fig. 1** Clustering of thermal relic neutrinos in the MW. **a** The neutrino contrast $\delta_\nu = n_\nu/\bar{n}_\nu - 1$ as a function of the distance $r$ to the galactic center of MW. The location of the Earth $r_\oplus = 8$ kpc is indicated by a gray vertical line. **b** The neutrino contrast $\delta_\nu$ as a function of the neutrino mass $m_\nu$ for three different distances of $r$ to the galactic center of MW. Both the horizontal and vertical axes are in logarithmic scale. Scatter points are the N-one-body simulation results. Each set of scatter points is well fitted by a power-law function $\delta_\nu \propto m_\nu^\gamma$

a given distance $r$, the relationship between $\delta_\nu$ and $m_\nu$ can also be well described by a power-law function. The obtained exponents $\gamma$ are also around two, so that the linear approximation[5,6] can be applied to this fully degenerate case as well.

**Implications on the direct detection of relic neutrinos.** Finally, we discuss the impact of gravitational clustering effects on the direct detection of relic neutrinos in the vicinity of the Earth. For concreteness, we take the PTOLEMY proposal as an example. In the PTOLEMY proposal, relic neutrinos are captured by the beta-decaying tritium nuclei $^3H$ via the process $\nu_e + {}^3H \rightarrow {}^3He + e^-$, and in the electron energy spetrum the signal events appear at a location that is $2m_\nu$ away from the beta-decay end point (assuming three degenerate neutrino masses). Although there exist many experimental issues[13,15], such as the limited energy resolution and the large background from beta decay, we here only discuss the overall capture rate, assuming a total mass of tritium of 100 grams as given in the PTOLEMY proposal (see the Methods section for the details on calculating the capture rate). Since the capture rate depends on the nature of neutrinos and the underlying assumptions on the thermal history of neutrinos, we here focus on the case of Dirac neutrinos and discuss three typical physics scenarios. Note that in the following discussions the left (right)-handed chiral states of neutrinos also mean the right(left)-handed chiral states of anti-neutrinos.

The Standard Case refers to the scenario predicted by the standard model of cosmology, i.e., only the left-handed chiral states of neutrinos were thermally produced in the early Universe, while for the right-handed chiral states of neutrinos their interactions with the particles in the thermal bath were so weak that their abundances can be safely neglected.

In the first NP scenario, i.e., NP Case I, the left-handed chiral states of neutrinos have the same thermal history as Standard Case. However, because of some NP effects, the right-handed chiral states of neutrinos can also be thermally produced in the early Universe[15,16], except for a higher decoupling temperature depending on the current cosmological constraints. Considering the current constraint on the extra effective number of neutrino species $\Delta N_{eff} < 0.53$ at the 95% confidence level (C.L.) by combining the Planck TT + lowP + BAO data sets with the helium abundance measurements[10], the number density of the right-handed chiral states is found to be at most 28% of that of the left-handed chiral states[15]. For illustration, we here set the number density of the right-handed chiral states to be 28% of that of the left-handed chiral states, when neutrinos began free-streaming.

The second NP scenario, i.e., NP Case II, is almost the same as NP Case I, except that the right-handed chiral states are non-thermal and their phase space distribution is assumed to be the fully-degenerate form[14]. To satisfy the same cosmological constraint as NP Case I, their number density can be 52% of that of the left-handed chiral states[17], when neutrinos began free-streaming.

In Fig. 3a we show the capture rate $\Gamma$ as a function of the total neutrino mass $\sum m_\nu \equiv m_1 + m_2 + m_3$ in Standard Case. Here $m_i$'s (for $i = 1$, 2, 3) are the three neutrino masses. The best-fit values of neutrino oscillation parameters from the latest global fit results[18] are adopted, and two neutrino mass hierarchies, normal hierarchy (NH) ($m_1 < m_2 < m_3$) and inverted hierarchy (IH) ($m_3 < m_1 < m_2$), are distinguished. Because of gravitational clustering effects, we find that if taking $\sum m_\nu = 0.23$ eV, the 95% C.L. upper limit allowed by the Planck TT + lowP + lensing + BAO + JLA + $H_0$ data sets[10], the capture rate can be enhanced by about 23% (31%) in the case of NH (IH), compared to the scenario without clustering. Because of the low target mass of tritium nuclei, distinguishing the neutrino mass hierarchies via the direct detection of relic neutrinos, given the current specifications of the PTOLEMY proposal[2], will be difficult. The capture rates in the NP Case I and II scenarios are given in Fig. 3b. Taking $\sum m_\nu = 0.23$ eV, the enhancement of capture rate due to gravitational clustering effects in NP Case I is the same as that in Standard Case, since in the two scenarios both the left-handed and right-handed chiral states of neutrinos possess the same thermal Fermi–Dirac phase space distribution. However, in NP Case II we find that the rate of enhancement turns out to be higher, 33% (45%) in NH (IH), due to the fact that larger clustering effects are observed in the fully-degenerate phase space distribution. Moreover, due to the clustering effects the differences between the capture rates in NP Case I and II (solid curves vs. dashed curves) become more prominent for larger values of $\sum m_\nu$. Therefore, gravitational clustering effects are useful for distinguishing NP scenarios involving different phase space distributions of relic neutrinos.

## Discussion

In this work, when performing the N-one-body simulation, we did not consider the impact of other nearby galaxies (or clusters) on the clustering of relic neutrinos in the MW. In fact, it was estimated that the Virgo cluster may create a neutrino over-density of the same order as that generated by the MW[8,12]. Therefore, we also need to include the gravitational potential of the Virgo cluster into simulation. With both the MW and the Virgo cluster, the shape of the gravitational potential is no longer

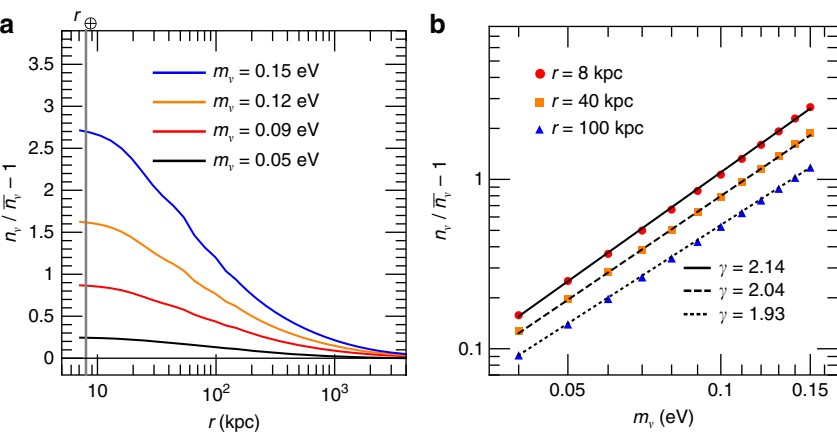

**Fig. 2** Clustering of fully degenerate relic neutrinos in the MW. The descriptions of **a**, **b** are the same as those in Fig. 1

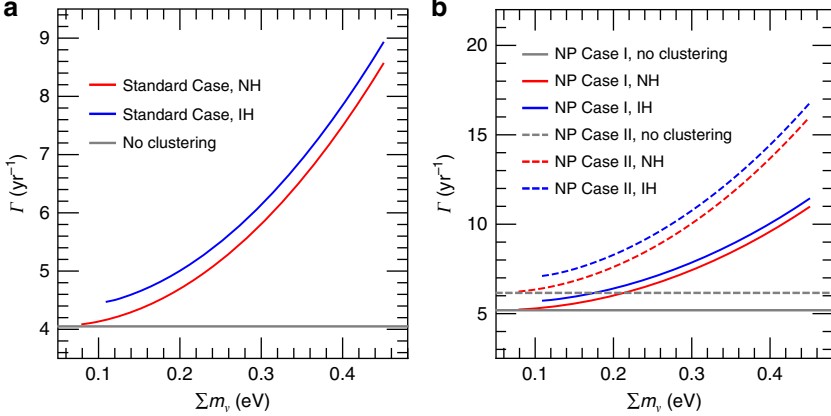

**Fig. 3** Capture rate $\Gamma$ in the PTOLEMY experiment. **a** Standard Case refers to the scenario predicted by the standard model of cosmology, i.e., only the left (right)-handed chiral states of (anti-)neutrinos were thermally produced in the early Universe. NH (IH) stands for the normal (inverted) mass hierarchy of neutrinos. **b** NP Case I and II are two new physics scenarios, and in both of them the thermal history of the left(right)-handed chiral states of (anti-)neutrinos is the same as Standard Case. In NP Case I the right(left)-handed chiral states of (anti-)neutrinos were also thermally produced in the early Universe, while in NP Case II the right(left)-handed chiral states of (anti-)neutrinos are non-thermal and possess the fully-degenerate phase space distribution. The number density of the right-handed chiral states of neutrinos is taken to be 28% (52%) of that of the left-handed chiral states of neutrinos in NP Case I (II), when neutrinos began free-streaming

spherically symmetric. However, since each relic neutrino still follows a one-body motion under the total gravitational potential, the N-one-body simulation and the reweighting technique are still applicable, except that the normalized evolution equations in Eq. (2) and the associated weight for each test particle in Eq. (3) would be in more general forms without the spherical symmetry. Moreover, the currently used matter density profile of the MW still has some uncertainties[8], and thus the reweighting technique would also become useful, were we plan to update the N-one-body simulation with more accurate matter density profile in the future. In short, the reweighting technique simplifies the task of using the N-one-body simulation method to evaluate the gravitational clustering effects on relic neutrinos, and is useful for studying the phenomenology associated with the future detection of the cosmic neutrino background.

## Methods

**DM and baryonic matter density profiles in the MW.** The DM density profile is taken to be a generalized NFW form[8],

$$\rho_{\rm DM}(r,z) = \mathcal{N}(z) \left[\frac{r}{r_{\rm s}(z)}\right]^{-\eta} \left[1 + \frac{r}{r_{\rm s}(z)}\right]^{-3+\eta}. \quad (7)$$

The parameter $\eta$ is kept to be redshift-independent, while $r_{\rm s}(z)$ and $\mathcal{N}(z)$ are evolved with redshift $z$. Here we adopt the best-fit values of $(\eta, r_{\rm s}(0), \mathcal{N}(0)) = (0.53, 20.29\ {\rm kpc}, 0.73)$ from a $\chi^2$-fit[8] to data[19]. The evolutions of $r_{\rm s}(z)$ and $\mathcal{N}(z)$ are dictated by the evolutions of the viral quantities $\Delta_{\rm vir}(z)$ and $c_{\rm vir}(M_{\rm vir}, z)$, which are defined as

$$\Delta_{\rm vir}(z) \equiv \frac{M_{\rm vir}}{\frac{4\pi}{3} a^3 r_{\rm vir}^3(z) \rho_{\rm crit}(z)}, \quad (8)$$

$$c_{\rm vir}(M_{\rm vir}, z) \equiv \frac{r_{\rm vir}(z)}{r_{\rm s}(z)}. \quad (9)$$

Here $M_{\rm vir}$ and $r_{\rm vir}(z)$ are the virial mass and radius, respectively, and they are related by the following equation

$$M_{\rm vir} = 4\pi a^3 \int_0^{r_{\rm vir}(z)} \rho_{\rm DM}(r', z) r'^2 {\rm d}r'. \quad (10)$$

Note that $M_{\rm vir}$ is redshift-independent. The evolution of the critical density $\rho_{\rm crit}(z)$ is

$$\rho_{\rm crit}(z) = \frac{3H_0^2}{8\pi G} \left[\Omega_{\rm m,0}(1+z)^3 + \left(1 - \Omega_{\rm m,0}\right)\right], \quad (11)$$

where $G$ is the gravitational constant, and $H_0$ and $\Omega_{\rm m,0}$ are the present values of the Hubble parameter and the matter density fraction, respectively. Here we adopt the best-fit values from the Planck data $(H_0, \Omega_{\rm m,0}) = (67.27\ {\rm km\ s^{-1}\ Mpc^{-1}}, 0.3156)$[10].

Given the three equations Eqs. (8), (9) and (10) and the quantities $M_{\rm vir}$, $\Delta_{\rm vir}(z)$ and $c_{\rm vir}(M_{\rm vir}, z)$, we can obtain the redshift evolution for $(r_{\rm s}, r_{\rm vir}, \mathcal{N})$. The evolution of $\Delta_{\rm vir}(z)$ is taken to be[20]

$$\Delta_{\rm vir}(z) = 18\pi^2 + 82\lambda(z) - 39\lambda(z)^2, \quad (12)$$

where $\lambda(z) = \Omega_{\rm m}(z) - 1$ with $\Omega_{\rm m}(z)$ being the matter density fraction at redshift $z$. With $\Delta_{\rm vir}(0)$ and Eqs. (8), (10) and (11), we obtain $M_{\rm vir} = 3.76 \times 10^{12} M_\odot$, being $M_\odot$ the solar mass. Subsequently, the evolution of $r_{\rm vir}(z)$ is also found from Eqs. (8) and (11). The evolution of $r_{\rm s}(z)$ is related to that of $r_{\rm vir}(z)$ via the concentration parameter $c_{\rm vir}(M_{\rm vir}, z)$. The evolution of $c_{\rm vir}(M_{\rm vir}, z)$ is assumed to be $c_{\rm vir}(M_{\rm vir}, z) = \beta c_{\rm vir}^{\rm avg}(M_{\rm vir}, z)$[8], where $c_{\rm vir}^{\rm avg}(M_{\rm vir}, z)$ is taken from N-body simulations[20],

$$\log_{10} c_{\rm vir}^{\rm avg}(M_{\rm vir}, z) = A(z) + B(z) \log_{10} \left[\frac{M_{\rm vir}}{1.49 \times 10^{12} M_\odot}\right], \quad (13)$$

with the functions of $A(z)$ and $B(z)$ given by[20]

$$A(z) = 0.537 + 0.488 \exp\left(-0.718 z^{1.08}\right),$$

$$B(z) = -0.097 + 0.024 z. \quad (14)$$

The value of $\beta = 2.09$ is then obtained from $c_{\rm vir}^{\rm avg}(M_{\rm vir}, 0)$ and $c_{\rm vir}(M_{\rm vir}, 0) = r_{\rm vir}(0)/r_{\rm s}(0)$. Finally, with the obtained $r_{\rm s}(z)$, the evolution of $\mathcal{N}(z)$ can be found from Eq. (10).

For the baryonic matter density profile, in reality it has a bulge shape for the central region of radius ~5 kpc[19] and extends to be a disc in the outer region. However, since the baryonic matter density is much smaller than the DM density in the disc region, we choose to ignore the disc part, and assume the overall profile of the baryonic matter density to be spherically symmetric. The actual baryonic matter density profile as a function of the radius at the present time is adopted from Fig. 2 in ref. [8]. The redshift dependence of baryonic matter profile is modeled with a normalization factor $\mathcal{N}_{\rm b}(z)$[8], and the evolution of $\mathcal{N}_{\rm b}(z)$ is found by averaging over the evolution of the total stellar mass as obtained in eight MW-sized simulated halos[21].

**Benchmark N-one-body simulation.** In the benchmark N-one-body simulation, we set $m_\nu = 0.15\ {\rm eV}$ and consider relic neutrinos with a Fermi–Dirac phase space distribution. The normalized evolution equations in Eq. (2) are used, and the scale factor is evolved as

$$\frac{{\rm d}a}{{\rm d}t} = H_0 \sqrt{a^{-1}\Omega_{\rm m,0} + a^2\left(1 - \Omega_{\rm m,0}\right)}. \quad (15)$$

We utilize an iteration procedure to discretize the initial phase space into finer parts[5,8], so that a smooth neutrino density profile can be achieved up to ~5 kpc. Because of spherical symmetry, the set of variables to be discretized is $(r, y, \psi)$. In general, we have $r \in [0, \infty)$, $y \in [0, \infty)$ and $\psi \in [0, \pi)$. However, for obtaining the

neutrino overdensity around the Earth, we can restrict the ranges for $(r, y, \psi)$. Specifically, we consider two regions of $r$, an inner region $R_I$: $r \in [0, 14.3 \text{ Mpc})$, and a outer region $R_O$: $r \in [14.3 \text{ Mpc}, r_{max}]$. The value of $r_{max}$ is taken to be the maximal radial distance that a neutrino with mass $m_\nu$ can travel, given that it must arrive at the Earth at $z = 0$ and with a momentum $p_{max}$. Here $p_{max}$ depends on the form of $f(y)$, e.g., for the thermal case we can truncate $y$ to $y_{max} = 10$, as the contribution from $y > 10$ is negligible. Although for $m_\nu = 0.15$ eV setting $r_{max} = 150$ Mpc is sufficient, we here take $r_{max} = 350$ Mpc in order to cover smaller neutrino masses.

In the inner region $R_I$ we consider $y \in [0, 10]$ and $\psi \in [0, \pi]$. In the coarse scan we set the resolution of $r$ to be $\Delta r = 14$ kpc, and divide the whole ranges of $y$ and $\psi$ into 120 parts evenly. After the coarse scan, we select the chunks that end up in the sphere of $r < r_7 = 7$ Mpc, and further divide the ranges of $r$, $y$ and $\psi$ into another 2, 4 and 4 parts, respectively, for the fine scan.

For the outer region $R_O$, we first notice from the results of ref. [5] that when $m_\nu \lesssim 0.3$ eV and $M_m \lesssim 10^{13} M_\odot$, with $M_m$ being the total mass of the gravitational source, the clustering effect of neutrinos is rather weak beyond the distance $r_7$. As a result, the initial direction of the momentum can be confined to $\psi(r) \in [\pi - r_7/r)$, as the test particles with $\psi(r) < \pi - r_7/r$ would never reach the location of the Earth. The range of $y$ in $R_O$ is also bounded as $y(r) \in [y_{min}(r), y_{max}(r)]$. The existence of $y_{min}$ is due to the fact that a minimum velocity is always needed in order to reach the location of the Earth, while for $y_{max}$ it is mainly because an escape velocity exists for a given gravitational potential. In the actual simulation, we obtain the dependence of $y_{min}$ and $y_{max}$ on $r$ through a quick numerical scan. Specifically, in the quick scan we vary $y$ within a wide range while having $\psi \in [\pi - r_7/r)$, and then find out the allowed range of $y$ that leads to the final value of $r$ below $r_7$. The lower and upper limits of such an allowed range of $y$ yield $y_{min}$ and $y_{max}$, respectively. We repeat this kind of numerical scan for a few values of $r$, and then perform a parametric fit to obtain the dependence of $y_{min}$ and $y_{max}$ on $r$. Having found the ranges of $y$ and $\psi$, we then perform a coarse scan with a resolution of $\Delta r = 140$ kpc and divisions of 120 parts for both $y$ and $\psi$. After the coarse scan, we again select the chunks that end up within the sphere of $r < r_7$, and then perform two rounds of fine scan. In the first round of fine scan, we further divide the ranges of $y$ and $\psi$ into another five parts for the selected chunks from the coarse scan. In the second round of fine scan we focus on the chunks that end up within the sphere of $r < 140$ kpc from the first round of fine scan, and another divisions of 20 parts are applied to both $y$ and $\psi$.

The last step of N-one-body simulation is to reconstruct the neutrino density profile from the discrete evolved samples. Adopting the kernel method[22] and with spherical symmetry, the neutrino number density profile can be estimated as

$$n_\nu(r) = \sum_{i=1}^{N} \frac{w_i}{d^3} K(r, r_i, d),$$ (16)

with the kernel function $K(r, r_i, d)$ given by

$$K(r, r_i, d) = \frac{1}{2(2\pi)^{3/2}} \frac{d^2}{r r_i} \left[ e^{-(r-r_i)^2/(2d^2)} - e^{-(r+r_i)^2/(2d^2)} \right].$$ (17)

Here $i$ is the sample index, $w_i$ ($r_i$) denotes the weight (final radial distance) of the sample $i$, and $N$ is the total number of samples. The window size $d$ plays a similar role as the bin size in the regular histogram. Therefore, too small values of $d$ would lead to large shot noise, while too large values would smooth out the detailed structure of interest. In this work we set $d = 8$ kpc, and find that varying $d$ by 3 kpc would modify the neutrino density $n_\nu(r_\oplus)$ by around 10%. Note that a similar level of uncertainty around 8% due to discrete sampling is also observed in ref. [8], and within the uncertainties our simulated results for the case of $m_\nu = 0.15$ eV agree with those in ref. [8].

**Capture rate in the PTOLEMY experiment.** The capture rate of relic neutrinos in the PTOLEMY experiment can be calculated as[13]

$$\Gamma \simeq \sum_{s_\nu = \pm 1/2} \sum_{j=1}^{3} \left| U_{ej} \right|^2 n_j(s_\nu) C(s_\nu) \overline{\sigma} N_T,$$ (18)

where $s_\nu$ denotes the helicity of neutrinos, $j$ labels the neutrino mass eigenstate, and $n_j(s_\nu)$ is the local number density of neutrinos with mass $m_j$ and helicity $s_\nu$. In addition, the total number of tritium nuclei is represented by $N_T$, and $\overline{\sigma} = 3.834 \times 10^{-45}$ cm$^2$ is the cross-section for the capture of relic neutrino on tritium nuclei. Since only the electron neutrinos can be captured by the tritium nuclei, we need to project each neutrino mass eigenstate into the electron neutrino flavor state, resulting in the factor $|U_{ej}|^2$ in the above formula. The lepton mixing parameters $|U_{ej}|^2$ are found by adopting the best-fit values of neutrino oscillation parameters[18], i.e., $(|U_{e1}|^2, |U_{e2}|^2, |U_{e3}|^2) = (0.665, 0.314, 0.022)$. The spin-dependent factor $C(s_\nu)$ is given by $C(s_\nu) = 1 - 2s_\nu v_{\nu_j}$ with $v_{\nu_j}$ being the velocity of neutrino with mass $m_j$. For the total neutrino mass $\sum m_\nu$ considered in Fig. 3, all neutrino mass eigenstates are non-relativistic at the present time, so that we can set $v_{\nu_j} \simeq 0$ and $C(s_\nu) \simeq 1$. As a result, the last quantity that we need to specify in Eq. (18) is the local number density of neutrinos $n_j(s_\nu)$, which depends on the underlying physics scenarios and the gravitational clustering effects.

In all the three physics scenarios considered in this work, the nature of Dirac neutrinos is assumed. Therefore, lepton number is conserved and thus only neutrinos, not anti-neutrinos, can be captured. Regarding the neutrino helicity states, in the early Universe both left-handed and right-handed chiral states are relativistic, and therefore they are left-handed and right-handed helical states as well. In the evolution of the Universe, the helicites of neutrinos are preserved, so that at the present time the left-handed and right-handed helical neutrino states correspond to the left-handed and right-handed chiral states in the early Universe, respectively. In Standard Case, the standard model of cosmology predicts the average number density of left-handed helical neutrino states to be $n_0 = 56$ cm$^{-3}$ for each mass eigenstate at the present time, while almost no existence of right-handed helical states. For a given value of the total neutrino mass $\sum m_\nu$, we can obtain the three individual neutrino masses with the two mass-squared differences from neutrino oscillation experiments. Here we adopt $\Delta m_{21}^2 \equiv m_2^2 - m_1^2 = 7.56 \times 10^{-5}$ eV$^2$ and $|\Delta m_{31}^2| \equiv |m_3^2 - m_1^2| = 2.55 \times 10^{-3}$ eV$^2$ from ref. [18]. Taking $\sum m_\nu = 0.23$ eV as an example, we obtain $(m_1, m_2, m_3) = (0.0711$ eV, $0.0717$ eV, $0.087$ eV) in NH. From the fitted function in Eq. (5) we find that the neutrino density contrasts $\delta_\nu$ are $(0.22, 0.23, 0.35)$ for the three mass eigenstates, respectively, and therefore the corresponding number densities of left-handed helical states around the Earth are 68.44, 68.64 and 75.53 cm$^{-3}$. Finally, from Eq. (18) we obtain the capture rate $\Gamma = 4.97$ yr$^{-1}$ in this case. The calculation of capture rates in NP Case I and II can be performed similarly, except that in NP Case I and II there are additional contributions from the right-handed helical states of neutrinos, whose average number densities at the present time are taken to be $0.28n_0$ and $0.52n_0$ for each mass eigenstate in NP Case I and II, respectively. Gravitational clustering effects on the right-handed helical states are calculated in the same way as the left-handed helical states, except that in NP Case II the fitted relation for the fully-degenerate phase space distribution should be used.

**Data availability**. The data that support the plots within this paper and other findings of this study are available from the corresponding author upon reasonable request.

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

## Acknowledgements

We are indebted to Zhi-zhong Xing for suggesting the topic and useful discussions. We also thank Ran Ding, Guo-yuan Huang, Yandong Liu and Shun Zhou for useful discussions. This work was supported by the National Natural Science Foundation of China (Grants No. 11522540 and No. 11690021), the National Program for Support of Top-notch Young Professionals, the Provincial Department of Education of Liaoning (Grant No. L2012087), and the China Postdoctoral Science Foundation (Grant No. 2017M610008).

## Author contributions

J.Z. and X.Z. jointly initiated the project. Both authors have contributed to this work.

## Additional information

**Competing interests:** The authors declare no competing interests.

