## [Peer Review File · Nature Communications]

Reviewers' comments:

Reviewer #1 (Remarks to the Authors):

The relic neutrinos from the Big Bang are predicted in the standard model of cosmology. The detection of such neutrinos will not only provide an independent support for the standard Big Bang theory, but also open a new window into the intrinsic properties of massive neutrinos. The PTOLEMY experiment is very promising to capture the Big Bang relic neutrinos by the beta-decaying tritium nuclei. With a nominal setup, PTOLEMY could observe the cosmological relic neutrinos at a rate of 4 events per year if neutrinos are Dirac particles, while 8 events per year for Majorana neutrinos. To calculate the event rate, it is very important to predict the local neutrino number density as precisely as possible. Since neutrinos are massive, the dark matter halo in the galaxy will change the local number density of neutrinos via gravity.

The manuscript contributes greatly to the study of gravitational clustering effects on relic neutrinos and presents a novel method to investigate these effects by numerical simulations. Although Ringwald and Wong did the first calculation in Ref.[4] long time ago, the authors have proposed a new reweighting approach, noticing that the equation of motion of a massive particle in the gravitational potential of the dark-matter halo does not actually depend on its mass. With this approach, one only needs to perform the detailed simulation once for a given neutrino mass and the distribution function, and then the results for other different masses and distribution functions can be obtained by using the previous simulation data and the reweighting factors. This increases significantly the efficiency of numerical calculations.

In addition, the manuscript is clearly written and the numerical results well support final conclusions. Hence I recommend the manuscript for publication in Nature Communications. However, the authors may like to improve the manuscript by addressing the following point:

The enhancement factor of local neutrino number density has been found to follow a power-law function of the neutrino mass with an index around two. It is pointed out in the manuscript that this index can be understood by using the linear perturbation theory. However, it should be better to recapitulate the underlying physics, and try to explain under which conditions the linear approximation is invalidated and the power-law index deviates from two.

Reviewer #2 (Remarks to the Author, see also attached file):

This paper explores the topic of neutrino clustering in the Milky Way gravitational potential. The authors use the N-one-body techniques already described in previous works, with the interesting addition of the reweighting technique which they develop. Such a method allows to reduce dramatically the computational time when exploring different values of the neutrino masses, thanks to the fact that the same benchmark N-one-body simulation can be used to compute several neutrino density profiles for different masses.

Despite the main content of the paper is surely interesting and provides an innovative technique, I find that some issues must be addressed before the article can be published.

General comments:

* is there a particular reason for which the reweighting technique is applied to the case of neutrino masses between 0.04 and 0.15 eV? Is it possible to apply it to even smaller masses, ideally down to 0.01 eV?

* The authors mention (pag. 5) that only $m=0.06$ and 0.15 eV are studied in the previous literature. Are the results found here in agreement with the previous ones? Since the method is the same as in ref. 9, I expect that for the benchmark run at 0.15 eV there is no difference, while

a comparison with the case 0.06 eV should measure the accuracy of the reweighting technique.

* when discussing the NP Cases I and II, the authors quote a number density of 28% or 52% for the right-handed states. Where do these numbers come from? How are they fixed? What is the total N_{eff} when adding the new contribution? is it compatible with cosmological constraints? The authors should at least briefly address these questions.

* In pag.11, the authors discuss how to possibly implement the clustering calculation when more than one object is included. I do not fully agree with their statements, although I may be wrong and only the full calculation can really clarify the subject. If you independently compute the neutrino clustering due to the different galaxies, you always start from the same initial condition of an isotropic and homogeneous neutrino number density, and evolve them towards the same galaxy. However, in the real case, neutrinos that may cluster towards one galaxy (if it is alone) can change their trajectory due to the presence of another galaxy/galaxy cluster. For example, when computing the overdensity due to the Milky Way, neutrinos that were initially coming from the direction of the Virgo cluster might not be able to reach our galaxy because of the gravitational attraction of the Virgo cluster. In other words, some of the neutrinos which were clustering around the MW will now cluster around Virgo and the clustering near Earth is not the sum of the two contributions as computed independently. If they agree, the authors may add a short explanation on this.

* In the Methods section, I think it should be fair to cite also the references from which the original data adopted in ref. 9 to describe the DM and baryons profiles were taken. Moreover, the shape of the baryon profile is not described at all, only its redshift evolution.

Comments on the text and exposition:

* I suggest a general spellcheck as there are many typos (nulei, kernal, osciallation, for example).

* The adopted lexicon is sometimes not very scientific. I discourage for example the use of "certain", as it is not suitable to describe with precision a method or some results.

* Many suggestions on how to improve the exposition may be found in the attached PDF. For example, in the first paragraph, I think it might be better to explain the difference between N-body and N-one-body simulations when the latter are first mentioned, and to add also there the citation to ref. 3, so that one can find immediately all the minimal information without needing to go to other references or scrolling the entire paper.

Some final comments on the abstract, that is the part of the publication which is more exposed to the public.

* there is a huge error, when saying that the number density contrast is *inversely* proportional to the square of the neutrino mass. Is the density contrast increasing for small masses? As it becomes clear reading the text, it is *directly* proportional to the square of the neutrino mass.

* a very vague limit (less than about 0.1 eV) is mentioned for the neutrino masses, without specifying if it refers to the single neutrino masses or to their sum. As a specific value is used in the text, why not using it here also? On the other hand, for the data combination mentioned in pag.9, reference 4 (doi:10.1051/0004-6361/201525830) reports a result of $\sum m_{\nu} < 0.21$ eV at 95% CL (eq. 54b): the authors should verify the number they are quoting. In any case, arxiv:1605.02985 contains more recent and stringent limits on the sum of the neutrino masses which may be (optionally) adopted.

* I do not find correct to say that the calculation of the clustering of relic neutrinos is "useful for the future detection of the cosmic neutrino background" (the same sentence appears in pag. 11):

the detection will be (possibly) obtained regardless of the clustering. It think that the message should be changed to say that it is interesting to study the clustering in order to gain more information from the event rate (Majorana/Dirac nature, to study the neutrino distribution, the galaxy content and other possible effects), for example saying "useful for studying the phenomenology associated with the future detection of the cosmic neutrino background".

Reply to Reviewer #1:

We would like to thank the referee for the careful and positive review. We are encouraged by the referee's comment that "The manuscript contributes greatly to the study of gravitational clustering effects on relic neutrinos". One suggestion has been made by the referee to which we respond below. In the revised manuscript we mark the corrections related to the referee's concerns in blue.

"The enhancement factor of local neutrino number density has been found to follow a power-law function of the neutrino mass with an index around two. It is pointed out in the manuscript that this index can be understood by using the linear perturbation theory. However, it should be better to recapitulate the underlying physics, and try to explain under which conditions the linear approximation is invalidated and the power-law index deviates from two."

In the revised manuscript, before Eq. (5) we add some discussions regarding the underlying physics of the linear approximation, and also discuss under what circumstance the linear approximation could break down so that the power-law index deviates from two.

Authors' Note:

To comply with the format requirements imposed by Nature Communications, some additional changes are made to the previous manuscript. The major ones include shortening the abstract and adding two additional paragraphs in the introduction (marked in magenta in the revised manuscript).

Reply to Reviewer #2:

We would like to thank the referee for the very detailed and careful review. Some important points have been raised by the referee to which we respond in detail below. In the revised manuscript we mark the corrections related to the referee's concerns in red.

General comments:

** is there a particular reason for which the reweighting technique is applied to the case of neutrino masses between 0.04 and 0.15 eV? Is it possible to apply it to even smaller masses, ideally down to 0.01 eV?*

The reasons why we consider the neutrino masses between 0.04 eV and 0.15 eV are as follows.

First, the consideration of 0.15 eV as an upper limit of the mass range in this work is mainly due to the fact that this value was commonly used in the previous studies (e.g., in ref. 9). Our investigation on this case can provide a direct comparison with the results derived in ref. 9. On the other hand, the value of 0.15 eV still roughly satisfy the current cosmological constraints from the Planck satellite mission. Here, 0.15 eV for a single neutrino mass would translate into a value of 0.45 eV for the total neutrino mass, which is still less than the 95% CL upper limit, 0.49 eV, derived by using the Planck TT, TE, EE + lowP data under the assumption of the Λ CDM cosmology (note that, the upper limit is 0.72 eV when using the Planck TT + lowP data; the upper limit is 0.23 eV when using the Planck TT + lowP + lensing + BAO + JLA + H0 data). Thus, the value of 0.15 eV considered in this work is roughly the highest value allowed by the current observations. (We note that, in the revised version, ref. 9 actually becomes ref. 8, which is because we have adjusted the order of the references in the revised manuscript.)

Second, for the lower limit of the mass range, 0.04 eV, it is mainly because when m_ν is below 0.04 eV, the obtained neutrino contrasts are less than 0.05 (also can be read from Figure 1), which means that the gravitational clustering effects are less than 5% for $m_\nu < 0.04$ eV. Such tiny clustering effects are hard to observe in the current setup of PTOLEMY. For example, given that the expected number of signal events is only 4 for the Dirac neutrinos, a rough estimation of the 1-sigma statistical uncertainty (without considering the background events) is about $\sqrt{4}/4 = 50\%$. Therefore, the gravitational clustering effects for very light neutrino masses can be neglected in the upcoming PTOLEMY experiment. We note that the exact value of neutrino mass below which the gravitational effects can be neglected depends on the considered experiments and also the possible uncertainties in those experiments. So before knowing the very details of those experiments, it is hard to make an accurate estimation on the value of neutrino mass for which the clustering effects are negligible.

The reweighting technique relies on the fact that relic neutrinos follow the one-body motion under the gravitational potential of dark matter and baryonic matter. So in principle it does not depend on the neutrino mass, and can be applied to even smaller neutrino masses. However, there is a technical issue in the N-one-body simulation, which has also been stated in the Methods section. Namely, in order to cover even smaller neutrino masses, the initially studied region indicated by r_{max} has to be enlarged, as neutrinos with smaller neutrino masses can travel farther.

In the revised manuscript, below Eq. (5) we add some discussions on the case of $m_\nu < 0.04$ eV, and comment that it would be hard for the current PTOLEMY to become sensitive to those tiny clustering effects.

** The authors mention (pag. 5) that only $m=0.06$ and 0.15 eV are studied in the previous literature. Are the results found here in agreement with the previous ones? Since the method is the same as in ref. 9, I expect that for the benchmark run at 0.15 eV there is no difference, while a comparison with the case 0.06 eV should measure the accuracy of the reweighting technique.*

Our results agree with those presented in ref. 9, after taking into account the uncertainties from discrete sampling, namely, the window size d in the kernel method of reconstructing the neutrino density profile.

Specifically, for $m_{\nu} = 0.15$ eV, the central value of $f_c (= n_{\nu}/n_{\nu}^0)$ in ref. 9 is around 2.4, while ours is about 2.15. So the difference is about 10%. However, there exist uncertainties coming from discrete sampling in both results. From Fig.4 in ref. 9 we know the uncertainty for the result in ref.9 is about 8%. In our case the uncertainty is about 10% (Note that on Page 19 of the previous manuscript we incorrectly stated that the uncertainty for the neutrino density contrast is 5%. In fact, it should be 20%. In the actual simulation we found the relative uncertainty of f_c to be about 10%. However, we made a conversion error in the previous manuscript when translating the relative uncertainty of f_c to that of density contrast δ_{ν} . Namely, since f_c is about twice of δ_{ν} in this case, so the relative uncertainty of δ_{ν} should be twice, instead of half, of that in f_c . Now on Page 18 of the revised manuscript we correct that error). So considering these uncertainties, our results are in agreement with those in ref. 9.

For $m_{\nu} = 0.06$ eV, even though we do not perform another independent simulation, we obtain f_c to be about 1.16 by using the reweighting technique, while the central value of the simulated result in ref. 9 is about 1.18. Considering that the uncertainty of the latter is also about 2% from Fig. 3 in ref. 9, both results then agree with each other, even without including the uncertainty from the error propagation from 0.15 eV to 0.06 eV in our case. Therefore, the reweighting technique is shown to be a valid method.

In the revised manuscript, we now add a few sentences below Eq. (5) to state that our results agree with those in ref. 9, after including the uncertainties. Moreover, on Page 18 in the Methods section we explicitly give the uncertainties in our results and those in ref. 9.

** when discussing the NP Cases I and II, the authors quote a number density of 28% or 52% for the right-handed states. Where do these numbers come from? How are they fixed? What is the total N_{eff} when adding the new contribution? is it compatible with cosmological constraints? The authors should at least briefly address these questions.*

On Page 10 of the revised manuscript, we now give the value of ΔN_{eff} explicitly, and also point out the references which provide the details of obtaining the numbers of “28%” and “52%” for the right-handed chiral states.

** In pag.11, the authors discuss how to possibly implement the clustering calculation when more than one object is included. I do not fully agree with their statements, although I may be wrong and only the full calculation can really clarify the subject. If you independently compute the neutrino clustering due to the different galaxies, you always start from the same initial condition of an isotropic and homogeneous neutrino number density, and evolve them towards the same galaxy. However, in the real case, neutrinos that may cluster towards one galaxy (if it is alone) can change their trajectory due to the presence of another galaxy/galaxy cluster. For example, when computing the overdensity due to the*

Milky Way, neutrinos that were initially coming from the direction of the Virgo cluster might not be able to reach our galaxy because of the gravitational attraction of the Virgo cluster. In other words, some of the neutrinos which were clustering around the MW will now cluster around Virgo and the clustering near Earth is not the sum of the two contributions as computed independently. If they agree, the authors may add a short explanation on this.

We agree that it is incorrect to say that the overall neutrino density can be obtained by considering the contributions from each galaxy first and adding them up later. In fact, although it is true that the Vlasov equation for the phase space distribution “ f ” is linear in the gravitational potential ϕ , it is not linear in both “ f ” and ϕ . There is a quadratic term involving both “ f ” and ϕ in the Vlasov equation. We overlooked that quadratic term and made wrong statements.

Now on Page 13 of the revised manuscript we remove those wrong statements. Instead, we explicitly say that the overall potential is no longer spherical when considering the gravitational potential of both MW and the Virgo cluster. However, we emphasize that the N-one-body simulation method and the reweighting technique are still applicable, since relic neutrinos still follow the one-body motion under the total gravitational potential, except that we may have to modify the normalized evolution equations and the weight formula to accommodate non-spherical symmetry.

** In the Methods section, I think it should be fair to cite also the references from which the original data adopted in ref. 9 to describe the DM and baryons profiles were taken. Moreover, the shape of the baryon profile is not described at all, only its redshift evolution.*

In the Methods section, the original references of obtaining the dark matter and baryonic matter are now added. Also, we add the details on the treatment of the shape of the baryon profile on Page 15.

Comments on the text and exposition:

** I suggest a general spellcheck as there are many typos (nulei, kernal, osciallation, for example).*

Typos are all corrected.

** The adopted lexicon is sometimes not very scientific. I discourage for example the use of "certain", as it is not suitable to describe with precision a method or some results.*

Inappropriate lexicons are revised.

** Many suggestions on how to improve the exposition may be found in the attached PDF. For example, in the first paragraph, I think it might be better to explain the difference between N-body and N-one-body simulations when the latter are first mentioned, and to add also there the citation to ref. 3, so that one can find immediately all the minimal information without needing to go to other references or scrolling the entire paper.*

Suggestions in the attached report are adopted. For example, in the first paragraph the reference to the N-one-body simulation is given in the first place, followed by a brief explanation on the differences between the N-body and N-one-body simulations.

Some final comments on the abstract:

** there is a huge error, when saying that the number density contrast is *inversely* proportional to the square of the neutrino mass. Is the density contrast increasing for small masses? As it becomes clear reading the text, it is *directly* proportional to the square of the neutrino mass.*

We thank the referee for pointing this out. Now in the abstract the error of “inversely proportional to” is corrected to “proportional to”.

** a very vague limit (less than about 0.1 eV) is mentioned for the neutrino masses, without specifying if it refers to the single neutrino masses or to their sum. As a specific value is used in the text, why not using it here also? On the other hand, for the data combination mentioned in pag.9, reference 4 (doi:10.1051/0004-6361/201525830) reports a result of $\sum m_\nu < 0.21$ eV at 95% CL (eq. 54b): the authors should verify the number they are quoting. In any case, arxiv:1605.02985 contains more recent and stringent limits on the sum of the neutrino masses which may be (optionally) adopted.*

The vague limit of “less than about 0.1 eV” is now removed in the abstract. Since according to the format requirements of Nature Communications the abstract cannot contain references, we decide not to mention a specific mass limit for neutrinos in the abstract. Actually, in the main text, we have used the upper limit of $\sum m_\nu < 0.23$ eV (on Page 11). We thank the referee for pointing out the mistake for the data sets used in this place, and we have corrected this error, i.e., “Planck TT + lowP + BAO data sets” is now replaced by “Planck TT + lowP + lensing + BAO + JLA + H₀ data sets”.

** I do not find correct to say that the calculation of the clustering of relic neutrinos is "useful for the future detection of the cosmic neutrino background" (the same sentence appears in pag. 11): the detection will be (possibly) obtained regardless of the clustering. It think that the message should be changed to say that it is interesting to study the clustering in order to gain more information from the event rate (Majorana/Dirac nature, to study the neutrino distribution, the galaxy content and other possible effects), for example saying "useful for studying the phenomenology associated with the future detection of the cosmic neutrino background".*

We agree with the referee’s point. The last sentence in the abstract ending with “... useful for the future detection of the cosmic neutrino background” (also on Page 11 of the previous manuscript) is modified to “... useful for studying the phenomenology associated with the future detection of the cosmic neutrino background”.

Authors’ Note:

To comply with the format requirements imposed by Nature Communications, some additional changes are made to the previous manuscript. The major ones include

shortening the abstract and adding two additional paragraphs in the introduction (marked in magenta in the revised manuscript).

Reviewers' comments:

Reviewer #1 (Remarks to the Author):

In the revised version, the authors have made proper changes in order to clarify the issue raised in the first report. Moreover, the manuscript has been greatly improved by addressing the questions from the other referee. As I already stated in the first report, the novel method suggested in the present manuscript will be very useful for the studies of cosmic neutrino background (CNB). The detection of CNB in future experiments will not only provide a strong support for the standard cosmology, but also open a new window to the intrinsic properties of massive neutrinos.

Therefore, I recommend it for publication in Nature Communications without any further changes.

Reviewer #2 (Remarks to the Author):

I am very happy that the authors properly considered and addressed all my previous comments. I think that the quality of the discussion increased enough to guarantee the publication.

Please let me comment a last thing.

In page 8, when saying that the clustering of neutrinos with a mass smaller than 0.04 eV would increase the number of events in PTOLEMY less than 5%, probably impossible to probe, the authors do not consider that PTOLEMY will not be able to detect relic neutrinos with such a small mass. A new PTOLEMY proposal is currently under development and as far as I know the most optimistic energy resolution will be around 0.05 eV, so that the minimum mass for which a CNB detection will be possible cannot be smaller than around 0.07 eV: this is completely independent of the neutrino clustering. Therefore, such small clustering effects are impossible to observe because they are out of the experimental possibilities.

Even in case of a plausible measure, however, I think other astrophysical uncertainties (the Virgo cluster contribution, for example) would dominate over such a small δ_{ν} . If the authors agree, I suggest to focus more on this aspect than on the PTOLEMY case when motivating the considered minimum mass of 0.04 eV.

Reply to Reviewer #2:

We would like to thank the referee for the helpful comment. Again, it helps us further improve the quality of the manuscript. In the revised manuscript we still mark the corrections in RED. Note that the colors marked in the previous revision are removed.

Referee's comment:

In page 8, when saying that the clustering of neutrinos with a mass smaller than 0.04 eV would increase the number of events in PTOLEMY less than 5%, probably impossible to probe, the authors do not consider that PTOLEMY will not be able to detect relic neutrinos with such a small mass. A new PTOLEMY proposal is currently under development and as far as I know the most optimistic energy resolution will be around 0.05 eV, so that the minimum mass for which a CNB detection will be possible cannot be smaller than around 0.07 eV: this is completely independent of the neutrino clustering. Therefore, such small clustering effects are impossible to observe because they are out of the experimental possibilities.

Even in case of a plausible measure, however, I think other astrophysical uncertainties (the Virgo cluster contribution, for example) would dominate over such a small δ_{ν} . If the authors agree, I suggest to focus more on this aspect than on the PTOLEMY case when motivating the considered minimum mass of 0.04 eV.

Our response:

We agree with the referee that due to the limited energy resolution the current PTOLEMY experiment cannot detect relic neutrinos with masses below 0.04 eV. We also agree that other astrophysical uncertainties may play more important roles in the case of such a small mass of neutrinos. On Page 8 of the revised manuscript we now add some discussions regarding both of the above points, and adopt them as the reasons of not considering neutrino masses below 0.04 eV.